# Nondestructive Evaluation of Lined Paintings by THz Pulsed Time-Domain Imaging

**Kaori Fukunaga**

National Institute of Information and Communications Technology, Tokyo 184-8795, Japan; kaori@nict.go.jp

**Abstract:** For the past 20 years, THz pulsed time-domain imaging has been used to study the internal structure of various paintings. The main advantage of this technique is that it can nondestructively provide depth profiles, as well as conditions of preparation and support layers, which are important in conservation planning. We have applied THz pulsed time-domain imaging to artworks with linings, which are additional layers of fabric or paper adhered to the back of an original painted work on canvas or paper to reinforce it, usually for conservation purposes. If the lining material or the interface between the lining and the original canvas or paper deteriorates over time, there is a risk of further problems. Therefore, THz pulsed time-domain imaging is expected to be suitable for examining lined paintings. In this paper, the investigation of artworks with lining layers using the THz pulsed time-domain imaging technique is presented, in addition to previous studies of Japanese panel screens and modern oil-canvas paintings with wax-resin linings, as well as a detached fresco painting mounted on a canvas by the "strappo" technique.

**Keywords:** terahertz; time-domain imaging; heritage science; paintings; lining

## 1. Introduction

Since the end of the 20th century, the use of terahertz pulsed time-domain imaging (THz pulsed TDI) has advanced [1]. Thanks to the development of transportable systems, it has been practically applied to the study of museum objects. In the beginning, the main goal of THz applications in cultural heritage was the spectroscopic imaging of painted surfaces [2–5]. These attempts were not of much interest to conservators. This is because most paintings are created with inorganic pigments obtained from minerals, which can be identified by well-established X-ray fluorescence and X-ray diffraction techniques, and multispectral imaging techniques from ultraviolet to near-infrared light can also be used for organic materials. Even if it is necessary to use the THz range for spectroscopy, a conventional FTIR system can cover the frequency range from 0.1 to 20 THz by adopting a far infrared option, so the need to use the pulsed time-domain system is not very high.

On the other hand, the time-of-flight technique with THz pulse waves that penetrate into opaque materials has a great advantage in the structural analysis of paintings. Although X-ray radiography is a technique widely used in the field of heritage science, information from all internal layers is superimposed in transmission mode, so it is not possible to obtain a cross-sectional image unless an X-ray CT system is available. Another well-established technique, microwave radar, has been used to detect buried archaeological objects and inspect buildings [6], but its resolution is not sufficiently high to observe paintings, owing to their long wavelengths. Using THz pulsed TDI, the internal structure of the painting can be observed layer by layer by detecting the reflection-pulses generated at the interfaces as the THz pulse propagates, and a cross-sectional image can be reconstructed from the output data. The internal structure of the paintings, particularly the condition of the preparation layer, is important to conservators, so small samples are taken and examined under the microscope when budget and time allow. The nondestructive observation of internal structure using THz pulsed TDI should therefore be of benefit to conservators as

well as art historians. With the development of portable systems in the early 21st century [7], it has begun to be practically applied to studying museum objects. Since the THz pulsed TDI technique was applied to one of the collections of the Uffizi Gallery in Florence, Italy, for the purpose of structural investigation [8], a large number of museum collections have been examined around the world [9–12]. Figure 1 shows examples of artworks observed by the National Institute of Information and Communications Technology.

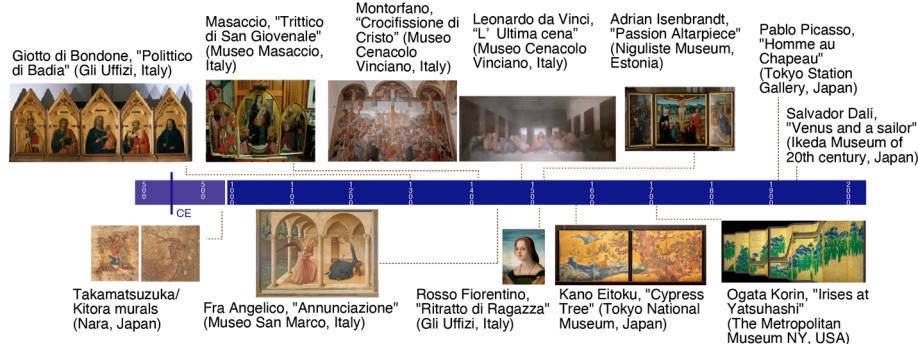

**Figure 1.** Examples of artworks observed by National Institute of Information and Communications Technology.

Various case studies published in the last decade have demonstrated the usefulness of THz pulsed TDI for the nondestructive observation of the internal layer structure of paintings, particularly preparation layers, which cannot be easily detected by other non-destructive methods using other electromagnetic frequency bands. THz pulsed TDI can provide not only depth profiles but also the condition of each internal layer, and research activities on its applications to heritage science have expanded to the point that it has even been included in a section of a general review paper on the THz technology itself [1]. Recently, attempts have been made to integrate other technologies such as infrared spectro-scopic imaging, optical coherent tomography (OCT) [13], and active thermography [14,15]. Signal analysis methods to improve the acquired image and to evaluate the obtained data have also been developed [16,17]. Applications have expanded beyond paintings to three-dimensional objects. For example, a system originally developed for the inspection of car paintwork, in which a sensor unit is mounted on an industrial robot, has been used to study sculptures and mummies [18]. There is also a project to observe ancient stone surfaces covered with lichen [19].

In this paper, we describe the application of THz pulsed TDI to the study of linings, which are not a part of the original painted work itself but are later attached to the back of the painting to reinforce it, not only for conservation purposes but also to replace fresco wall paintings and to incorporate artwork into furniture. Since the lining is an additional support for the existing artwork, its deterioration reduces the robustness of the entire work. Observation of the structure and condition of the lining by THz pulsed TDI can provide useful information for conservation planning.

## 2. THz Time-Domain Imaging Technique

Figure 2 shows a schematic of internal structure observation by THz pulsed TDI. When a THz pulse is applied to a two-layer object, as shown in Figure 2a, the first reflection pulse is generated at the surface. THz pulses penetrate most dielectric materials, and the second reflection pulse is generated when the THz pulse reaches internal interface A. The third reflection occurs at internal interface B. Then, the signal passes through to the air. As shown on the right-hand side of Figure 2a, the reflection pulse sequence appears with a delay determined by the depth from the surface and the propagation velocity. This so-called time-of-flight method is widely used in various other nondestructive test techniques, such as the ultrasonic method.

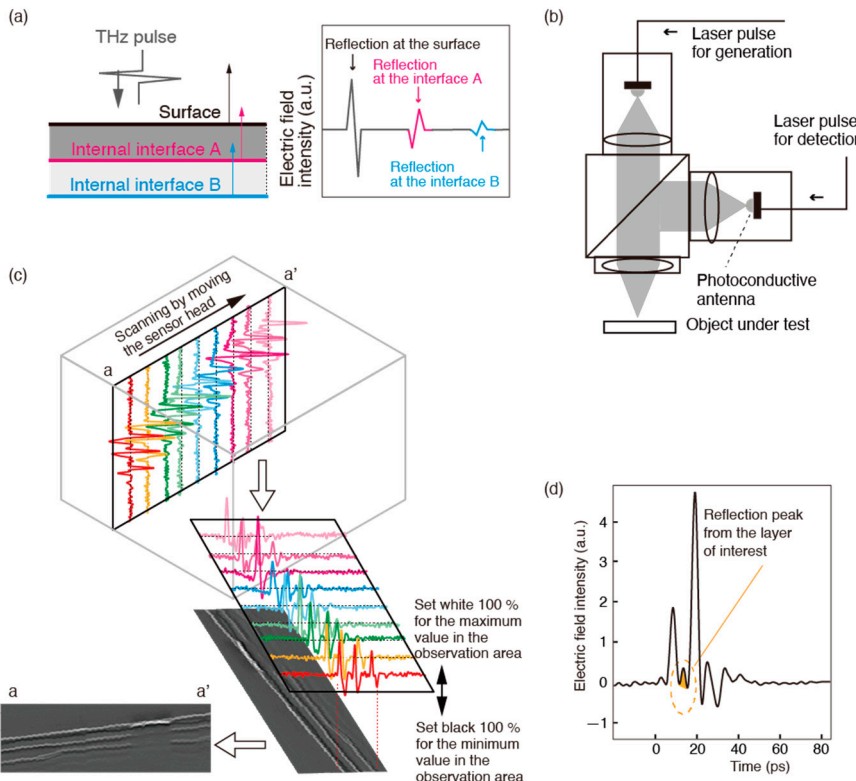

**Figure 2.** Schematic of THz pulsed TDI: (**a**) model of the reflection pulse sequence, (**b**) sensing unit of a commercial system, (**c**) cross-sectional image obtained by scanning, (**d**) extraction of the reflection pulse at given depth.

Because the THz frequency range is lower than the range of light and higher than that of radio waves, the THz pulse can penetrate opaque objects, and the resolution in the lateral direction (approximately 0.2 mm in the best case with a transportable system) is sufficiently high to examine objects made by human hands. By scanning an object along a line using the sensor unit (Figure 2b), which consists of a source, a mirror, and a detector, a cross-sectional image along this line can be obtained. As shown in Figure 2c, each waveform of the output signal represents the depth profile at that point as a sequence of pulses. By moving the sensor unit along the line marked a–a′ using a scanning system, a series of depth profiles can be obtained. Then, by setting the maximum value to be white and the minimum value to be black, and by displaying the intensity as a difference in brightness, a cross-sectional image along the line a–a′ is obtained in greyscale. The experimental results presented in this paper are all shown after deconvolution. An area image of the sample can be obtained by placing values at each measurement point. These values can be calculated by integrating the whole signal or a particular reflection pulse. As shown in Figure 2d, the state of the layer of interest can be obtained by extracting a reflection pulse at a specific delay time.

We have been using a portable scanner type THz pulsed TDI system in museums for more than 10 years. The results presented in this paper were obtained using the three commercial systems shown in Figure 3: T-Ray 4000 and T-Ray 5000 (Luna Innovations Inc., Ann Arbor, MI, USA) [7,20], and a THz scanner [21] (Pioneer Corporation, Saitama 350-8555, Japan). The name of the instrument is indicated in the caption of each experimental result. In our experience, observations are possible up to 5 mm in depth from the surface with a resolution of about 0.3 mm. The measurement time for scanning an area of 30 cm$^2$ at 1 mm steps in the lateral direction is approximately 30 min with the latest T-Ray 5000. Judging from the experience of observing a large number of artworks using transportable THz pulsed TDI systems available on the market, layers of painting material consisting mainly of fine inorganic mineral grains and organic binders can be observed with a resolution of about 0.3 mm in the axial direction down to a depth of about 5 mm. According to

several conservators, it is of practical use for observing the preparation layers of classical paintings and lining layers, as it provides information from deeper below the surface than microscopic examination of samples taken from the painting generally does.

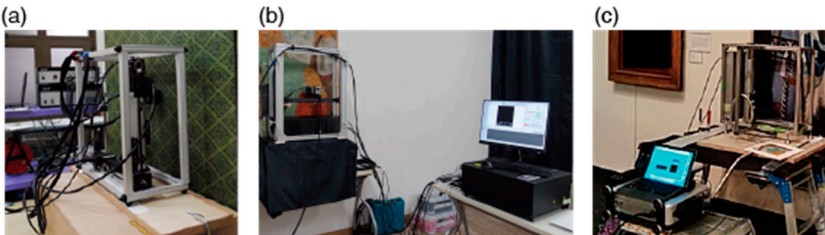

**Figure 3.** Examples of transportable THz TDI systems: (**a**) T-Ray 4000 (Luna Innovations Inc., Ann Arbor, MI, USA), (**b**) T-Ray 5000 (Luna Innovations Inc., Ann Arbor, MI, USA), (**c**) THz scanner (Pioneer corporation, Saitama 350-8555, Japan).

Note that there are various factors that affect THz pulsed TDI. First, THz waves, a frequency band of nonionising electromagnetic waves, are completely reflected by metal surfaces, so the area underneath cannot be observed from the metallic side. Even if no metal is present, surface conditions, including inclination and uneven surfaces, can cause scattering, thereby preventing the propagation of the incident THz wave into the desired object. Similar phenomena also occur inside inhomogeneous materials such as coarse fibres and aggregates mixed into calcium compounds used for preparation layers. For example, details of difficulties in THz pulsed TDI are explained in [22] (pp. 56–62).

### 3. THz Time-Domain Imaging in Heritage Science

THz pulsed TDI was first applied to the study of Renaissance panel paintings, a typical structure of which is shown in Figure 4. Here, it is important to choose the best frequency in accordance with the depth from the surface; classical techniques, such as X-ray radiography and photographic techniques do not allow the observation of preparation layers. Experimental results obtained by THz pulsed TDI include cross-sectional images at given positions, as well as sectional images at given depths. These images reveal the conditions of the substrate, the plaster layer, and the interface between them. Indeed, being able to look at a cross-sectional image without extracting small real samples can be useful for conservators.

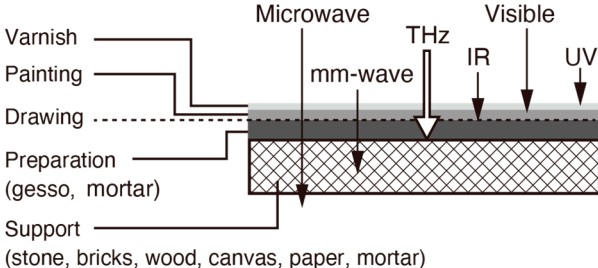

**Figure 4.** Typical structure of a painting and the frequency band suitable for observing each layer.

Figure 5 shows a typical cross-sectional image of *Polittico di Badia* (1301–1302) by Giotto di Bondone, obtained nondestructively by THz pulsed TDI, as well as area images at each depth. Unlike modern tempera panel paintings, which have a layer of gesso on a sheet of fabric directly adhered to the panel with animal glue, there are two layers of gesso with a sheet of fabric sandwiched between them. Here, the sheet of fabric is used to ease the tension caused by the deformation of the wood panel owing to changes in temperature and humidity. According to historians, such "double layers of gesso" mean that the artist followed a medieval procedure in creating this panel painting. The regular dotted pattern of the fabric is observed in Figure 5b, and the uneven wood surface in Figure 5a suggests

that the wood was carved with a knife, and the tool marks are observed as linear radial patterns in Figure 5c. The complete scientific analysis and conservation process are reported in [8], a work edited by historians.

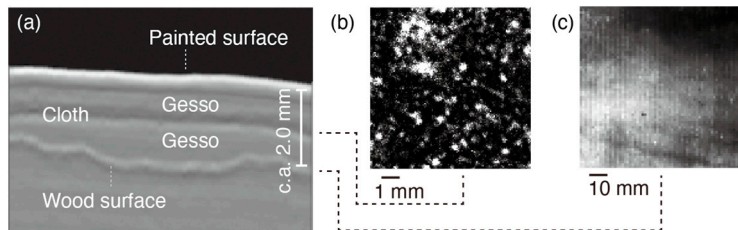

**Figure 5.** Examples of THz imaging results of *Polittico di Badia* by Giotto: (**a**) cross-sectional image, (**b**) sliced image at the interface between two gesso layers, (**c**) sliced image at the surface of the wood panel. (Measurement system: T-Ray 4000, Luna Innovations Inc., Ann Arbor, MI, USA).

Another example of the advantages of THz pulsed TDI is revealed by comparison with transmission X-ray radiography. Figure 6 shows the visible and radiographic images of part of the *Passion Altarpiece* (1510–1520) by Adriaen Isenbrant, which has been repainted several times over time [23]. Historians have suggested that the current coat of arms might have been painted over another older one, judging from the appearance of an invisible round shape in the X-ray image, as shown in Figure 6a,b. We observed the same area by THz pulsed TDI. The cross-sectional image along the dotted line shown in Figure 6c revealed that only the area of the coat of arms has two layers, and that a simple figure exists on the surface of the bottom layer, as shown in Figure 6d, at approximately 0.7 mm below the surface. On the basis of these results, historians have suggested that there is a house mark under the present coat of arms [24]. From these experimental results, it is clear that the entire structure and condition of each internal layer of a panel painting can be observed in a nondestructive and non-contact manner by THz pulsed TDI, thus contributing to conservation planning.

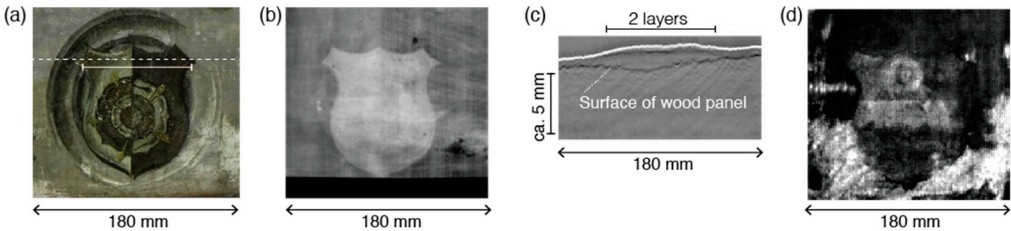

**Figure 6.** Examples of THz imaging results of *Passion altarpiece* by Adriaen Isenbrandt: (**a**) visible image, (**b**) X-ray radiography, (**c**) cross-sectional image along the dotted line, (**d**) invisible mark appeared below the coat of arms. (Measurement system: THz imaging scanner, Pioneer corporation, Saitama 350-8555, Japan).

The application of THz pulsed TDI has been extended to the study of canvas paintings and even mural paintings. In the case of frescoes, thick preparatory layers of lime containing coarse and/or fine sand are applied over the supporting stones or bricks. Internal scattering by the sand and surface irregularities affect the propagation of the THz pulse. As a result, the information obtained is often limited to the vicinity of the surface, and a clear layer structure such as that of a panel painting cannot be observed. The THz pulsed TDI has been applied to investigate the internal structures of masterpieces on walls, such as *Annunciation* (1440–1445) by Beato Angelico, *Crucifixion of Jesus* (1495) by Donato Montorfano, and *The Last Supper* (1495–1498) by Leonardo da Vinci. The results revealed the depth of cracks visible on the surface and the layered structure of repainted areas [25,26].

## 4. Artworks with Lining

Lining is a technique in which a piece of canvas or paper is attached to the back of an original work painted on canvas or paper to reinforce it, usually for conservation purposes. If the lining material or the interface between the lining and the original canvas or paper deteriorates over time, there is a risk that not only the appearance but also the physical condition of the artwork, including its support, will deteriorate. A typical thickness of lining layers is approximately 0.3–0.5 mm, and it is expected that these layers will be easily distinguished by THz pulsed TDI.

### 4.1. Oil Paintings

Lining is used for canvas paintings to reinforce the original paintings, using adhesive [27,28]. For centuries, animal glue, which can be removed and replaced when the lining layers deteriorate, has been used. In the mid-19th century, a new technique using a mixture of beeswax and some types of resin was developed in the Netherlands. In this process, wax resin melted at high temperatures penetrates the original canvas and lining fabric, which are then pressed together to form a solid thick canvas. The painted surface of the original artwork itself is subjected to high temperatures and high pressures. Worse still, in methods invented in the USA, the original canvas and lining fabric together with wax-resin are mounted on a metal plate, or a metal foil sheet is inserted between the layers of lining fabric [29,30]. Although the wax-resin lining is a highly invasive technique, it was considered to be a permanent solution, and spread around the world in the 1970s. Subsequently, problems such as deformation and decolouration have been recognised by conservators [31]. Thus, in the 21st century, wax-resin lining is no longer recommended. However, many artworks, particularly modern oil paintings, are lined with wax-resin, and continue to deteriorate. The study of the lining layers and interfaces is of great practical importance in planning the reconservation of existing artworks.

We previously examined the oil painting *Homme au chapeau* (1915) by Pablo Picasso. This artwork, which was painted on very thin canvas, was mounted on a new lining canvas with a wax-resin composite during the previous conservation in the USA. According to historians, the original artwork itself has been painted over at least once [22,32]. Figure 7 shows the THz reflection image and its cross-sectional image along the line indicated by the three arrows. Figure 7a suggests that this observation area can be divided into three parts A, B, and C, as indicated by the dotted lines, and the cross-sectional image along the three arrows (Figure 7b) clearly indicates the interface between the lining canvas and the thin original thin canvas.

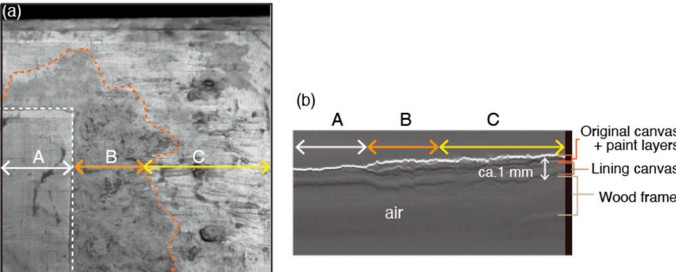

**Figure 7.** An example of THz imaging results of *Homme au chapeau* by Pablo Picasso: (**a**) THz reflection image, the visible image of which cannot be shown due to copyright, (**b**) cross-sectional image along the arrows in (**a**). (Measurement system: T-Ray 4000, Luna Innovations Inc., Ann Arbor, MI, USA).

The rectangular part A is considered to be the original painting, and the thickness of the paint layer is estimated to be less than 0.2 mm, as the painting and original canvas layers cannot be distinguished in this case. There is an additional layer of paint in parts B and C, and another additional layer is observed in part B.

Another example is the oil painting *Venus and Sailor* (1925) by Salvador Dalí, which was mounted on an aluminium honeycomb panel using a wax-resin composite. The artist

himself said that he painted over it several times. As can be seen in all the cross-sectional images along a–a′, b–b′, c–c′, A–A′, B–B′, and C–C′ (Figure 8), extremely uniform lining layers are commonly observed, so it is assumed that the set of these layers consists of wax-resin impregnated fabric. In addition, the number of paint layers varies from place to place, so Dalí may not have removed the old paint when he applied new paints. Although extremely high thermal energy had to be applied to the painting in order to mount it on a metal plate with high thermal conductivity, such treatment was previously considered to be superior [33].

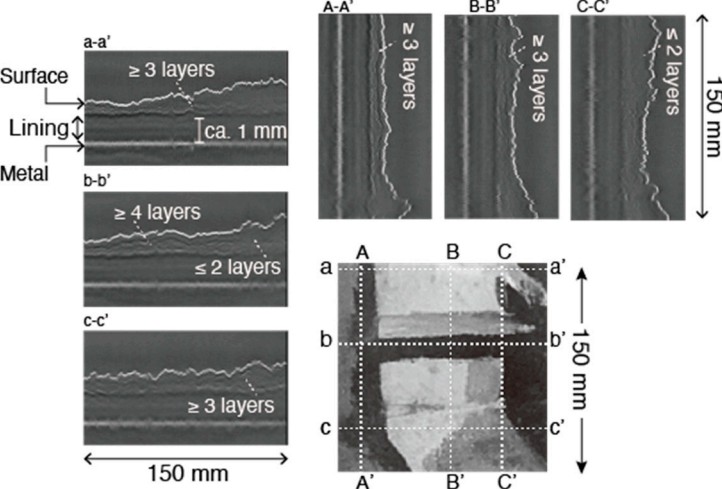

**Figure 8.** Examples of cross-sectional images of *Venus and Sailor* by Salvador Dalí, indicated with an infrared image of a part of the painting. The visible image cannot be shown due to copyright. (Measurement system: T-Ray 4000, Luna Innovations Inc., Ann Arbor, MI, USA).

We made canvas painting samples with wax-resin linings of different adhesion strengths and found that the lining condition affects the painting surface itself [15]. At present, we have confirmed that uneven areas can be clearly observed by THz pulsed TDI and also by active pulse thermography. Futhremore, it was possible to distinguish whether the thick internal layer was air or solid, or animal glue or wax-resin, from the cross-sectional image. These results will be published in the near future.

### 4.2. Panel Screeens

Traditional Japanese paintings are often displayed as part of the furnishings of a room, such as panel screens and sliding doors. In such cases, the painting itself is painted on a thin sheet of paper and lined with a set of several sheets of paper mounted on a wooden lattice, using animal glue, as shown in Figure 9.

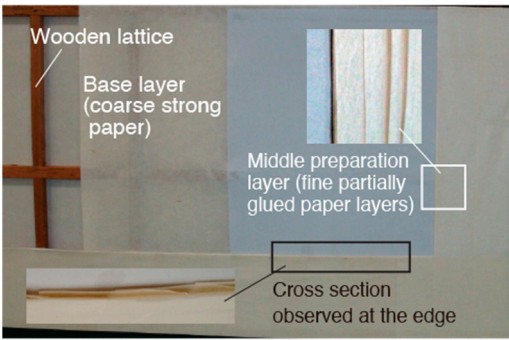

**Figure 9.** Model of support with preparation layers of a traditional Japanese panel screen. The painted paper is mounted on the support.

In a full-scale restoration, the entire support is removed and replaced with a new set of lining layers. If the internal condition of the lining layers could be known in advance, it would be possible to estimate the skill and time (i.e., cost) required to replace the support.

*Cypress Trees* (1590) by Eitoku Kano was originally painted as a fusuma-e (sliding door) and later transformed into a panel screen. The catches of the doors were removed, leaving holes in the centre of the painting. It is expected that these holes were specially treated on the reverse side, presumably with additional pieces of paper; their internal structure was revealed by THz pulsed TDI. Figure 10a shows one of the observation areas containing part of a catch and its cross-sectional image along the dotted line. As can be seen from the cross-sectional image, the hole made by removing the catch has been patched with gilded paper and the edge has a layer of paint on top of the gold. The number of sheets of paper underneath varies from place to place. Figure 10b shows the extracted image of the layer immediately below the painted paper. A complex arrangement of rectangular pieces of paper is clearly observed. These pieces may have been added to cover the holes of the catches. During the conservation treatment of panel screens, all support materials are removed, allowing conservators to verify the images obtained by THz pulsed TDI. As shown in Figure 10c, these results were found to be in good agreement [22] (pp. 135–138).

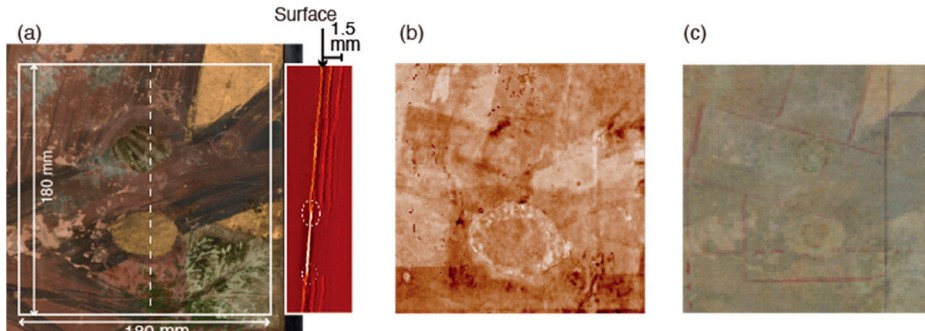

**Figure 10.** Examples THz imaging of *Cypress Trees* by Eitoku Kano: (**a**) observation area indicated in visible image and cross-sectional image, (**b**) area image extracted at the surface of the preparation layer, (**c**) a sketch by a conservator during the treatment. (Measurement system: T-Ray 4000, Luna Innovations Inc., Ann Arbor, MI, USA).

### 4.3. Fresco Painting after Detachment

To protect wall paintings from the destruction of the buildings themselves or the deterioration of their surroundings, or to display them in other places such as museums, it is necessary to remove the painted surface from the wall, and if necessary, to mount it on another support. The method of removing the entire supporting layer of lime mortar, known as "arriccio", has been used since Roman times. There is a risk of damage to the painted surface owing to mechanical stress. During the Renaissance period, the demand for the removal of wall paintings had increased as a result of urban reconstruction, and a new technique, "stacco", whereby only the "intonaco" layer is removed, was developed. The authors applied THz pulsed TDI to observe the layer structure of fragments of East Asian wall paintings removed by the stacco technique, and compared the results with the visible image at the edge. The conditions of the upper preparation layer, such as the depth of cracks, and the interface between the upper and lower preparation layers were clearly observed. However, the internal condition of the lower preparation layer could not be observed because the aggregate fibre used in the lower preparation layer of Asian mural paintings is coarse and interferes with the propagation of the THz pulse [22] (pp. 104–107). On the other hand, in the strappo technique invented in the 18th century, only the painted surface (i.e., the layer of calcite with the embedded pigments) is removed. First, cheesecloth, a very thin and strong linen fabric, is applied to the surface of the painting with animal glue. When the cheesecloth is removed from the wall surface after the animal glue has completely dried, only the painted surface layer is removed from the wall and transferred

to the cheesecloth side. The cheesecloth is then mounted onto a lining canvas using a water-insoluble adhesive. After the painted layer has completely adhered to the new lining canvas, the remaining cheesecloth is removed with warm water. The animal glue softens in warm water and the cheesecloth can be removed without damaging the painted surface. Contemporary fresco artists often use the strappo technique because then the works can be exhibited in museums in the same way as canvas paintings. Ideally, in the strappo process of a buon fresco painting, only the surface calcite layer of pigments is removed from the wall, leaving the intonaco and arriccio layers on the wall. However, if there are defects such as delamination between the intonaco and arriccio layers, the intonaco layer may not be able to withstand the strength of the glue and may detach from the wall. In addition, if the glue is not washed off completely after being applied to the lining canvas, the residual glue on the surface may shrink as it dries, causing the paint layer to peel off as if it were being lifted from the surface.

Since there is no previous work on the THz pulsed TDI of a strappo painting, it is worthwhile to measure a typical and ideal strappo painting. Figure 11a shows a reproduction of Michelangelo's *The Libyan Sibyl* (16th century CE) in the Sistine Chapel by the modern fresco artist Sachiko Okada. Figure 11b shows the image obtained by integrating the total reflected wave and the cross-sectional images along the indicated lines. The layer structure is almost uniform, and the thicknesses of the paint layer and the lining canvas part are almost the same regardless of the location. There is a small area where the lining canvas is visible owing to missing paint, as indicated by a dotted square. The area has low reflectance, resulting in a dark spot, and the cross-sectional image also clearly shows discontinuity, as seen in Figure 11c. The surface of the lining canvas was estimated by THz pulsed TDI to be 0.5 mm from the surface, which agrees with the actual measurements of the painting.

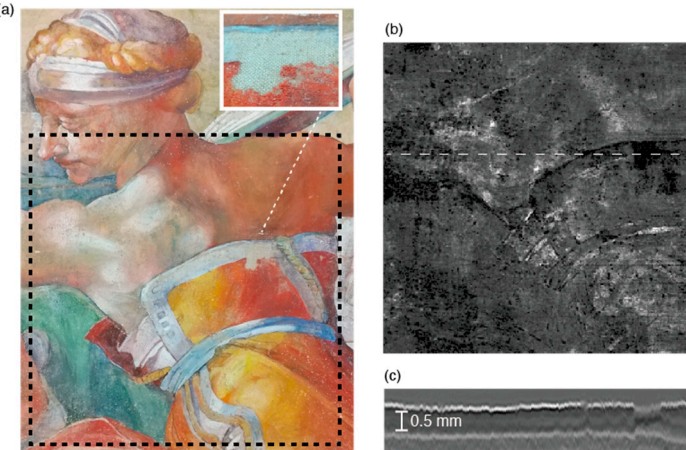

**Figure 11.** THz imaging of a strappo painting, a reproduction of Michelangelo's *The Libyan Sibyl* in the Sistine Chapel by Sachiko Okada: (**a**) Observation area in the visible image, and a detail of the missing part, (**b**) THz reflection image, (**c**) cross-sectional image along the white dotted line in (**b**). (Measurement system: T-Ray 5000, Luna Innovations Inc., Ann Arbor, MI, USA).

The work *Memories of Italy* (1958) was painted by Loka Hasegawa (also called Luca Hasegawa) on his return from Italy in 1958 by using the buon fresco technique and framed using the strappo process. There are many defects on the surface of the painting, and it appears to be in need of conservation treatment, so we observed the internal condition for future conservation planning. The surface of the artwork is not flat, as can be seen from a photograph taken with raking light (Figure 12a). As the maximum scanning area of the system used in this study is 30 cm × 30 cm, *Memories of Italy* was scanned in small regions, as shown in Figure 12b. Here, the position of the system was manually adjusted each time, so that the THz irradiation and detection conditions between regions. Thus, details of the internal structure of the painting in two characteristic regions, C1 and B4, are described.

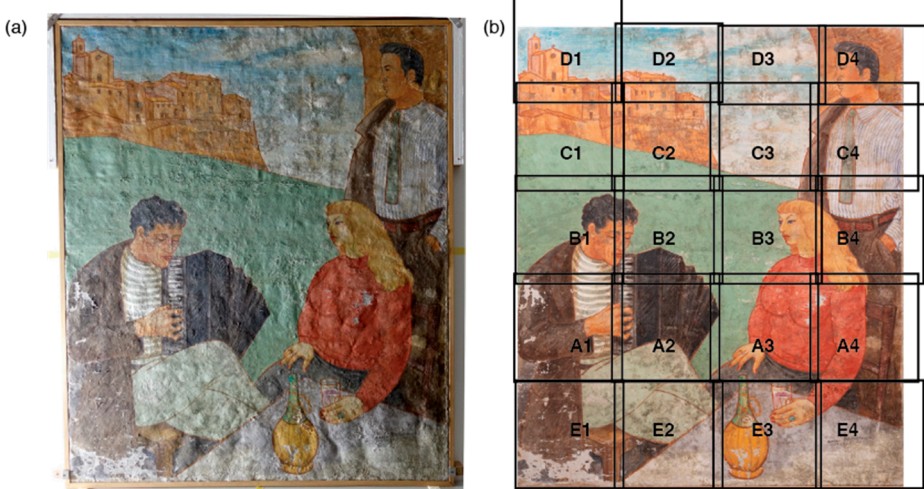

**Figure 12.** Photographs of *Memories of Italy* by Loka Hsegawa: (**a**) photograph taken with raking light, (**b**) observation regions divided for scanning.

Figure 13a shows the visible image of region C1 taken with raking light. Although the surface is uneven, the cross-sectional images (Figure 13b) are relatively uniform, and the thickness is almost constant, similar to those of the reproduction of *The Libyan Sibyl*. The area images at the surface of the painting and at the estimated area of the lining layer are shown in Figure 13c,d, respectively. Only small surface irregularities affect the area image at the surface of the painting, and the cloth grain of the lining canvas can be seen in the image extracted at the internal interface. Therefore, the strappo process was successful in the C1 region.

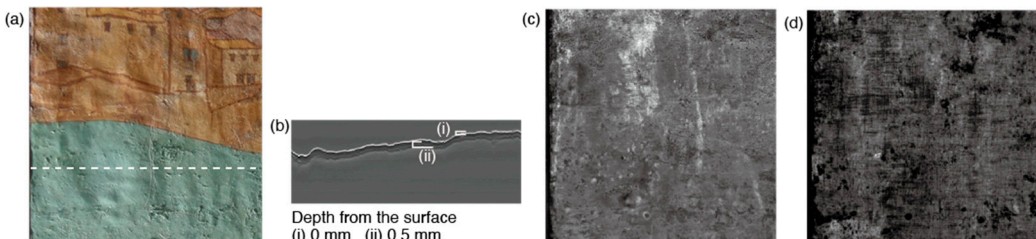

**Figure 13.** THz imaging of the region C1: (**a**) photograph taken with raking light, (**b**) cross-sectional image along the dotted line indicated in (**a**), (**c**) THz area image at the surface, (**d**) THz area image extracted at the depth of 0.5 mm from the surface. (Measurement system: T-Ray 5000, Luna Innovations Inc., Ann Arbor, MI, USA).

In order to discuss the cross-sectional image along the line across the joint, the visible image of area B4 taken with raking light is shown rotated by 90 degrees in Figure 14a. The cross-sectional images along the dotted line in area B4 vary considerably from place to place, and particularly thick layers are observed, as shown in Figure 14b. The sliced image at a depth of 0.5 mm to 1 mm from the surface, where the lining fabric is present, is shown in Figure 14c. The thick layer is not uniform, so it is unlikely to be an adhesive material such as plaster or cement used to attach the layer of paint to the lining canvas. It is reasonable to assume that this thick layer is residual mortar that has been stripped from the wall. The complex structure of the layers indicates that the strappo treatment was not successful in this area. There is clearly a joint of lining canvas sheets, as shown in the enlarged view in Figure 14a. We expected to see two layers, similar to the paper sheets in the panel screen. However, although the area is particularly thick, the internal structure is not sufficiently clear to distinguish between two sheets. This is because the THz pulse was attenuated owing to surface irregularities and scattering in the lime layer with coarse sand.

This could be verified from the reverse side if the artwork will be undergoing conservation in the future.

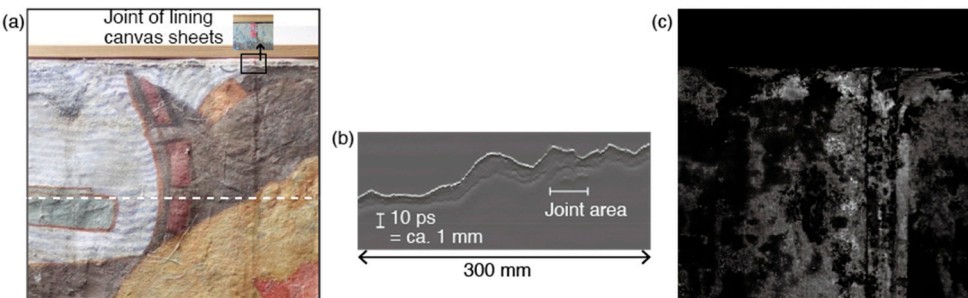

**Figure 14.** THz imaging of the region B4: (**a**) photograph taken with raking light, rotated 90 degrees, (**b**) cross-sectional image along the dotted line indicated in (**a**), (**c**) THz area image extracted at the depth from 0.5 mm to 1 mm from the surface. (Measurement system: T-Ray 5000, Luna Innovations Inc., Ann Arbor, MI, USA).

## 5. Conclusions

While cultural heritage research generally focuses on the original work itself, the history of past treatments is also extremely important for conservation planning. In particular, the deterioration of irreversible linings, such as wax-resin and strappo, affects the value of the original artwork. Observation of the internal structure using THz pulsed TDI should contribute to the examination not only of the original artworks, but also of irreversibly lined artworks, and could eventually provide useful information to help conservators develop better lining methods as well as post-conservation examinations.

The usefulness of THz pulsed TDI has been demonstrated in many previous studies, but it has not yet been widely used in the heritage science community, compared with material identification and mapping by X-ray fluorescence and multispectral imaging from ultraviolet to infrared regions. The reasons for the lack of THz pulsed TDI users in heritage science are that there are still few affordable and transportable devices, but essentially, there is a lack of information being communicated to historians and conservators. Although the argument that "interdisciplinary collaboration is important for innovation" has been popular since the 1980s, few countries have institutions where historians, conservators, and scientists work together. In many countries, including Japan, it is extremely difficult to establish a cultural project with an adequate budget in science and technology organisations. As a result, technical information is limited to the personal connections of THz experts if they publish their work in 'regular' conferences and journals. In Europe, conferences related to cultural heritage are more active than in other parts of the world, and engineers of X-ray fluorescence and multispectral imaging, as well as OCT manufacturers, actively promote their technologies to, and even codevelop specialised equipment for, cultural heritage research with historians and conservators. In the case of THz pulsed TDI, several manufacturers have recently started to present applications to heritage science mainly for public relations purposes, and this situation is bound to improve. We hope that companies and/or research institutions that offer measurement services at an affordable price will emerge, so that in the future, THz pulsed TDI will become a common tool in cultural heritage research.

**Funding:** This research received no external funding.

**Data Availability Statement:** The data presented in this study are available in the article.

**Acknowledgments:** The author would like to thank colleagues included in the referenced papers, and is grateful to Sanae Kita of the Fujisawa City Art Gallery for giving the opportunity to examine *Memories of Italy* by Loka Hasegawa, and to the artist Sachiko Okada for providing us with the reproduction of Michelangelo's *The Libyan Sibyl*.

**Conflicts of Interest:** The author declare no conflict of interest.

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
