# Peer review of "Nondestructive Evaluation of Lined Paintings by THz Pulsed Time-Domain Imaging"

_heritage, doi:10.3390/heritage6040183_

Round 1
Reviewer 1 Report
The actuality of the problem is important. The objects of investigation are masterpieces, so it is very interesting. Authors are big expert in investigation museum objects by terahertz techniques, so experiment, methods are good described. The only two things are missed. First one is very poor list of references, they missed terahertz experts like group of Patrick Mounaix, Paris group with Vincent Detalle, many italian groups, who working on investigation of terahertz for museum objects for many years. The second thing is very bad written Conclusion. It might be said that Conclusion is missed at all. Minor mistakes are: 1) Fig 15 is too dark and it is not clear still for what authors put them in article, 2) "each internal layer А of panel painting" - 120 line, A is extra.
Author Response
First of all, I would like to thank the reviewer for the encouraging comments and suggestions. I attached the corrections in PDF file.

Reviewer 2 Report
Manuscript ID: heritage-2184951
Type of manuscript: Article
Title: Nondestructive evaluation of various lined paintings by THz pulsed time-domain imaging for their conservation planning
1. Recommendation:
Accept after minor revisions
2. Comments to the authors:
2.1 Overview and general recommendation
The paper "Nondestructive evaluation of various lined paintings by THz pulsed time-domain imaging for their conservation planning" presents an interesting application of the THz imaging technique to the study of lined paintings. Its application can be useful to monitor the effectiveness and stability of the treatment. However, the paper requires some improvements to increase the clarity and to better support the data presented. For this reason, I recommend reconsidering after major revisions. I explained in detail my concerns below.
2.2 Major comments
The paper requires English editing and revision. It is also necessary to revise the technical terminology. Moreover, there is no Materials and Method section and the technical description of the instrumentation used in the studies presented is missing.
As the title mentions, a key element of this work is the lining technique, yet there is not much information regarding this procedure, therefore, adding some information regarding the lining treatment (e.g., when and why it is applied, in general, what are the techniques?) should be added to the introduction section. This is important since you studied some examples with a particular technique, wax resin, which is a very invasive method used in the past and is not used anymore in European countries but only in some areas, for example, Mexico. Can this technique be applied to other lining methods? Additionally, the resin-wax lining is applied when paint layers are damaged and fragile, thus it is also a consolidation method to stabilize the paint layers, can the THz technique evaluate the consolidation effect?
Finally, the paper would certainly benefit from a more detailed revision on the application of the THz imaging technique to the study of paintings (generally focused on the study of paint layers) in the Introduction, to highlight the main contribution of this work (investigation of the interface with the support and the lining of paintings and detached wall paintings). Also, adding further discussion to the conclusions could help, for example, explaining the main contributions, such as the possibility to understand mechanical failure and monitor the adequate adhesion from the support to the lining.
2.3 Minor comments
- Revise the journal guidelines, for example, Figure citations in the text should be Figure # instead of Fig. #.
- Keywords. Consider adding “paintings” and “heritage science” to the keywords to increase the findability of the article.
- Page 1, 1. Introduction, Lines 24-25. Please add a brief explanation of what THz are and their spectral range to help non-specialized readers to better understand the technique.
- Page 2. Section 2. Line 51. Please move Figure 2 to its first mention in the text.
- Page 2. Line 72. You mention a maximum depth penetration of 5 mm, is this enough to observe the complete stratigraphy – including the lining layer – of a painting? Also, does this penetration depth is influenced by the chemical composition of the layers (e.g., metallic leaves or other materials)? Please elaborate a little bit more about the limitations of the technique.
- Page 2. Line 72. Please specify if you are talking of axial or lateral resolution.
- Page 3. Line 93. Artwork titles should be written in italics without quotation marks. Also please add the complete information for example the author (mentioned only in the Figure caption) and the date or period. This should also be for the other paintings mentioned in the article.
- Please revise the technical terminology. For example, on page 5, line 131 “board painting” can be substituted by “panel painting”.
- Page 5, Section 4.1 Lines 156-157. It is not clear if the lining procedure was performed by Picasso or by a conservator. Please revise the text to make clearer this information. If necessary, add a citation to the main source of information.
- Page 8, Section 4.3, Line 212. Please change the subtitle to “Fresco painting after detachment” since you mention more than one technique of detachment (strappo and stacco).
- Page 8. Line 235. “melts” is not an adequate term. Please revise.
- Page 10. Figure 11. Please indicate the line from which the cross-sectional view was obtained also in a) to make it clearer.
- Please explain better all the elements (a,b,c..) in the Figure captions.
- Some of the images are small and thus hard to understand. Also, adding scale bars to the images is needed to better understand the dimensions of the studied areas.
Author Response

(The authors gave the same response as above.)

Reviewer 3 Report
This paper is very well written but needs a better and more comprehensive conclusion.
Also there needs to be a sense of the future work as to where all this can be applicable in heritage science apart from paintings. The authors need to provide a sense of where this research work is heading in the future.

Author Response

(The authors gave the same response as above.)

Round 2
Reviewer 2 Report
I want to thank the author for considering and integrating the previous comments.
The paper has been improved considerably and the author's response is very clear. Therefore, I suggest accepting the article after some minor revisions. I explain in detail below.
Page 1. Lines 26-27. "This is because most paintings are created pigments from inorganic minerals" probably the phrase can be rewritten as: "This is because most paintings are created with inorganic pigments obtained from minerals".
Page 1. Lines 34-44. When talking about studying internal layers in paintings, X-ray radiography is a technique widely employed in the field of Heritage Science. However, the THz technique has the advantage that it can give cross-sectional information, overcoming the problem of the superposition of layers in 2D X-ray images. Consider adding a brief line on this to highlight the advantage of the THz technique.
Author Response
Dear editors and reviewers,
Thank you so much for your encouragements.
I attached my responses to each of the comments (in bold italics) from the reviewer.
